# Environmental adversity is associated with lower investment in collective actions

**N. Lettinga**[1☯]*, **P. O. Jacquet**[1,2☯], **J-B. André**[2], **N. Baumand**[2], **C. Chevallier**[1]*

**1** Laboratoire de Neurosciences Cognitives et Computationnelles (LNC[2]), Département d'Études Cognitives, INSERM, École Normale Supérieure, PSL Research University, Paris, France, **2** Institut Jean-Nicod, Département d'Études Cognitives, INSERM, École Normale Supérieure, PSL Research University, Paris, France

☯ These authors contributed equally to this work.
* niels.lettinga@ens.fr (NL); coralie.chevallier@ens.fr (CC)

## Abstract

Environmental adversity is associated with a wide range of biological outcomes and behaviors that seem to fulfill a need to favor immediate over long-term benefits. Adversity is also associated with decreased investment in cooperation, which is defined as a long-term strategy. Beyond establishing the correlation between adversity and cooperation, the channel through which this relationship arises remains unclear. We propose that this relationship is mediated by a present bias at the psychological level, which is embodied in the reproduction-maintenance trade-off at the biological level. We report two pre-registered studies applying structural equation models to test this relationship on large-scale datasets (the European Values Study and the World Values Survey). The present study replicates existing research linking adverse environments (both in childhood and in adulthood) with decreased investment in adult cooperation and finds that this association is indeed mediated by variations in individuals' reproduction-maintenance trade-off.

**Data Availability Statement:** The datasets, pre-registrations and R scripts used in this study are available through the Open Science Framework: https://osf.io/642hd/ for the European Values Study

## 1. Introduction

Environmental adversity is often conceptualized as the product of two different dimensions: harshness (i.e., externally caused levels of morbidity-mortality that an individual cannot control) and unpredictability (i.e., spatial-temporal variation in harshness) [1]. In humans, adversity is closely aligned with economic deprivation and resource scarcity [2–4] and is commonly associated with a wide range of detrimental biological outcomes and behaviors, ranging from reduced lifespan [5], detrimental health behaviors [6–10], suboptimal financial decisions or lower education [11–17]. Adversity has also been targeted as a potential candidate causing variability in a number of social behaviors seemingly unrelated to those mentioned above, such as social preferences [18–21], social learning biases [22–26], social trust [27–29] and cooperation [30]. However, whether cooperation is decreased or increased in adverse environments is still debated.

In theory, the relationship between environmental adversity and cooperation could go either way [30]. On the one hand, adverse environments might be associated with increased

and https://osf.io/37bs2/ for the World Values Survey.

**Funding:** This study was supported by the Institut d'Etudes Cognitives (ANR-10-IDEX-0001-02 FrontCog), the Institut national de la santé et de la recherche médicale (INSERM), an ERC consolidator grant (no. 8657) and an "Action Incitative" funding from the École Normale Supérieure. The funders had no role in study design, data collection and analysis, decision to publish, or preparation of the manuscript.

**Competing interests:** The authors have declared that no competing interests exist.

cooperation as a means to mitigate direct survival threats, such as pooling material resources to reduce the impact of external events like famine [31,32]. On the other hand, adverse environments might be associated with decreased cooperation because adversity constrains people to focus on trying to meet their individual needs [30].

Whether adverse environments are associated with decreased or increased cooperation seems particularly influenced by the way cooperation is measured. The main distinction is between self-reported questionnaires and field experiments on the one hand and behavioral experiments (i.e., economic games) on the other hand. Self-reported measures of cooperation and economic games differ in several important ways: (1) self-reports are more influenced by social desirability bias, (2) they are based on people's limited insight on their own behavior and (3) self-reports are based on attitudes while economic games are based on actual behavior. It is important to note that some of the studies cited as support for the association between adverse environments and increased cooperation [31,32] use agent-based models, which are distinct from self-reported measures, field experiments and behavioral experiments.

The existing empirical evidence is currently inconclusive. When cooperation is measured via self-reported questionnaires or in field experiments, studies usually report an association between adverse environments and decreased cooperation. Notably, Korndörfer et al. [33] and Schmukle et al. [34] found a negative correlation between lower social class and cooperative behaviors (e.g., donating, volunteering, helping) using questionnaire data from large and representative international samples (average N over 9 studies = 20.243). Using field experiments, Andreoni et al. [35] and Nettle et al. [36] also found that lower social class correlates with decreased cooperation.

When cooperation is measured using behavioral experiments (i.e., economic games and actual behavior in various settings under scrutiny of the experimenter), the results are much more mixed. Some studies report an association between adverse environments and decreased cooperation [33,34,36–38], but other studies find the opposite pattern [39–41] or no effect at all [42,43].

The reason why these behavioral experiments give mixed results is unclear, but several explanations have been put forward. First, some of these studies have small sample sizes, which may explain, for example, why Piff et al. [41], who found a positive relation between environmental adversity and cooperation, was not replicated in a high-powered pre-registered study [42]. Second, these studies usually only use one economic game, which may lower their generalizability [44]. In accordance with this idea, McAuliffe et al. [45] found that single economic games do not correlate with real-life cooperation, but a general factor based on several economic games does. Finally, there is increasing evidence that economic games lack ecological validity [46].

Beyond establishing the correlation between adversity and cooperation, the channel through which this relationship arises also remains unclear. In a large review paper, Pepper and Nettle [47] put forward the idea that there might be a common origin to the constellation of behaviors observed in adverse environments, which is that they reflect a contextually appropriate need to favor immediate over long-term benefits. Recent studies indeed reveal that people living in adverse conditions prefer immediate rewards over delayed rewards, discount future rewards more, are more pessimistic about their future, and are more impulsive than those living in more affluent conditions [48–53]. In sum, people's living environment appears to affect the present-future trade-off, with adverse conditions promoting present-oriented behaviors, and affluent conditions promoting future-oriented behaviors.

Evolutionary models [54,55] and empirical evidence [56–60] suggest that variations in cooperation might also depend on variations in the present-future trade-off, and ultimately on the environmental factors that shape it. Cooperation is defined as "a behavior that provides a

benefit to another individual (recipient), and the evolution of which has been dependent on its beneficial effect for the recipient" [61,62]. Cooperation has two subcategories: altruistic and mutually beneficial behaviors [62]. We are particularly interested in mutually beneficial cooperative behaviors, where the actor incurs immediate costs and the benefits are delayed. In this framework, cooperation is a future-oriented strategy [63,64]: in the short term, it is more advantageous to reap immediate, smaller, but more certain benefits by cooperating less but in the long term, it is more advantageous to invest in large-scale cooperation so as to reap longer-term benefits such as increased social reputation.

The way the present-future trade-off is instantiated at the psychological level might partly result from how it is embodied at the biological level [47,65], notably through the way organisms optimally allocate their limited stock of energy between reproductive goals and maintenance goals [1,65–68]. Similarly to the present-future trade-off observed at the psychological level, the shape of the reproduction-maintenance trade-off is partly conditioned by the environment, and therefore subjected to plasticity during an individual's life. In line with this phenotypic plasticity perspective, correlational research in humans living in industrial and post-industrial societies suggests that experiencing adversity during childhood accelerates reproduction while decreasing investment in health [69–71]. By contrast, safe childhood conditions are associated with delayed reproduction [1,66,72–75] and are negatively associated with a number of morbidity risks [76]. In sum, prioritizing reproduction is classically viewed as a present-oriented life-history strategy because it comes with direct and indirect survival costs that might otherwise be minimized by delaying reproduction. The reason is that adverse conditions make life uncertain, so that one cannot be sure that an investment made in the present (e.g., health efforts) will pay-off in the future (e.g., living longer). Counteracting these risks requires prioritizing immediate but certain benefits (i.e., reproduction) at the expense of long-term but uncertain benefits (i.e., somatic investments and health) [77–80].

Taken together, these observations raise the possibility that the effect of environmental adversity on cooperation might be conditional on the shape of the reproduction-maintenance trade-off. More specifically, individuals who experienced environmental adversity–both during childhood and adulthood–might invest less in cooperative activities, and this effect might be conditioned at least partly on the priority they give to reproduction at the expense of health. Decreasing the costs of cooperative interactions might be contextually adaptive in the most deprived societies where social trust is lower and interpersonal violence is higher, and where people feel more exposed to risks of exploitation outside their kinship network [81–83].

In this paper, we test the relationship between childhood and current environmental adversity and people's propensity to invest in collective actions, which we use as a proxy for cooperation. Collective action can indeed be defined as "multiple individuals pay the cost of contributing to a group project and share in the resulting benefit" [84,85]. We further test the hypothesis that the relationship between adversity and collective action is conditional on an individuals' reproduction-maintenance trade-off. Our specific hypotheses were that (1) childhood environmental adversity is associated with lower investment in collective actions, and that this effect is mediated by variations in individuals' reproduction-maintenance trade-off; (2) current environmental adversity is associated with lower investment in collective actions, and that this effect is mediated by variations in individuals' reproduction-maintenance trade-off.

To test these hypotheses we applied multivariate models (structural equation modeling) on two independent datasets containing information about the values, the beliefs and the behaviors of more than 295.000 participants representative of more than 100 countries all over the world (the European Values Study and the World Values Survey). The robustness of the models was assessed by employing a discovery/ replication procedure. First, each dataset was

randomly split into two subsamples of similar size: a discovery sample and a replication sample. Second, the discovery sample was used to design and test the model. We then pre-registered the analysis plan in the Open Science Framework. Finally, the replication sample was used to check the robustness of the model. The present study replicates existing research linking adverse environments in childhood and adulthood with decreased investment in adult cooperation [33,34,36–38], and finds that this association is mediated by variations in individuals' reproduction-maintenance-trade-off. Furthermore, it adds to the consistent pattern of results (i.e., a relationship between adverse environments and decreased cooperation) found by existing studies that use self-reported questionnaires to measure cooperation [33,34].

## 2. Study 1: Childhood adversity and collective action in the European Values Study

We tested the association between childhood environmental adversity, the reproduction-maintenance trade-off and adult investment in collective actions by applying structural equation modeling on the European Values Study dataset [86]. The European Values Study has relied on social scientists since 1981 to collect data from large representative samples of the populations from 47 European countries. In this study, we used the fourth wave of the survey for which data was collected between 2008 and 2010. We focused on the fourth wave because it includes the same collective action variables as the World Values Survey, which gives us the opportunity to replicate our initial findings on an independent sample.

### 2.1 Methods

**2.1.1 Respondents.** 66.281 respondents living in 46 different countries are included in wave 4 of the European Values Study. However, some respondents could not be included in our analyses. Two of the variables used to model the reproduction-maintenance trade-off were only relevant for respondents with children, which led us to focus on the 45.624 respondents who report having children. We randomly split this sample into a discovery subsample (22.570 respondents) and a replication subsample (23.054 respondents). In both subsamples, we excluded respondents with too much missing data (the criterion was set to more than two standard deviations from a respondent's respective subsample mean) which led to 20.755 respondents for the discovery sample and 21.156 respondents for the replication sample. The final discovery sample after listwise deletions included 16.790 respondents (females: N = 9.837; mean age = 52 ± 15 sd; mean number of respondents per country = 382 ± 126 sd) and the final replication sample after listwise deletions included 17.087 respondents (females: N = 9.957; mean age = 52 ± 15 sd; mean number of respondents per country = 380 ± 136 sd). The final replication sample after listwise deletions consists of 45 countries (descriptive statistics for each country are detailed in S1 Table).

**2.1.2 Variables.** We selected variables tapping three constructs: childhood environmental adversity, the reproduction-maintenance trade-off and collective action. Childhood environmental adversity was modeled as a formative latent variable including the following items: "death of the father before age 16" (scale 0–1), "death of the mother before age 16" (scale 0–1), "parental education" (scale 1–8: the higher the score the lower the educational level of the parents), "parents had problems replacing things" (scale 1–4: the higher the score the more difficulty replacing things) and "parents had problems making ends meet" (scale 1–4: the higher the score the more difficulty making ends meet). Based on the recommendations of Brumbach et al. [69], childhood environmental adversity was modeled as a formative latent variable rather than a reflective latent variable. This was done because adverse childhood events can be seen as risk factors that are not necessarily correlated with one another. However, they all

contribute to the cumulative probability of developing a particular outcome (in our case an increased investment in reproduction or somatic maintenance). For example, having been exposed to the death of a parent, lower parental education or lower parental socioeconomic status are three events that increase the probability of experiencing an adverse childhood environment, but that might occur independently.

The reproduction-maintenance-trade-off was modeled by constructing a reflective latent variable based on "number of children", "age at first birth" (calculated from the birth year of the respondent and the reported birth year of their first child) and "subjective health status" (scale 1–5: the higher the score the better the respondents' reported health). These items were chosen because they approximate the reproduction-maintenance trade-off [1]. For example, recent studies report that some health costs are induced by increased investment in reproductive functions positively associated to fertility–in both women (e.g., high number of pregnancies, high parity) and men (e.g., high level of testosterone). These health costs include increased production of pro-inflammatory cytokines, insulin resistance, increased production of triglycerides and LDL cholesterol, increased oxidative stress, altered immune function, cardio-vascular diseases [66,75,87] and have long-lasting health correlates, including on people's lifespan, which has been shown to be well captured by self-reported health [88].

Cooperation was modeled by constructing a reflective latent variable. In articles using the European Values Study and the World Values Survey [89,90], cooperation is often assessed using questions tapping cooperative morality (e.g., the respondent's opinion about claiming government benefits to which you are not entitled; avoiding a fare on public transport; cheating on taxes if you have a chance). Such questions do not test actual cooperation, but rather people's moral judgments. Importantly, moral judgments are not systematically aligned with actual behavior. For instance, individuals who claim that it is wrong to cheat on taxes may actually cheat themselves. Individuals living in low trust societies are indeed more likely to cheat because they think that others are cheating [91] and they are also more likely to be in favor of harsh punishments for cheaters because they do not trust that cheating can be avoided by means other than harsh punishments [37].

To address this problem, we chose to focus on respondents' actual investment in collective actions. We scanned the European Values Study to identify variables that had previously been found to successfully predict real-life cooperation [33] and that tapped actual behaviors involving concrete costs and delayed benefits for the agent and benefits for others. Two sets of variables met our a priori defined criteria: how much someone volunteers (scale 0–15) and how often someone participates in political actions that might benefit the community, without receiving financial compensation (scale 0–5) (items are detailed in S2 Table). People who volunteer or take part in political actions indeed incur concrete costs, including time and energy, to engage in actions that benefit others. Furthermore, volunteering and political actions also have delayed benefits for the actor. Volunteering, for example, is positively associated with concern for social reputation or image [92] and is a means of obtaining human capital that will yield returns in the labor market [93,94]. Political action, for example voting, has also been shown to correlate with higher social reputation [95,96]. Although volunteering and political action are limited in their own way (for instance, many cooperative interactions happen outside the activities described in these items), these items have the advantage of being the most concrete instances of true investment in collective goods available in the European Values Study. Volunteering was calculated by taking the sum of 15 separate items that indicate if respondents volunteer for different groups (e.g., religious organizations, trade unions and sport groups). Political action was calculated by taking the sum of five separate items that indicate if respondents take part in different political actions (e.g., signing a petition, joining boycotts and attending lawful demonstrations).

**2.1.3 Model specification and fit.** First, the reproduction-maintenance trade-off was regressed on childhood environmental adversity. Second, collective action was regressed on the reproduction-maintenance trade-off and childhood environmental adversity. Third, childhood environmental adversity was modeled as a formative latent variable. These variables need to be scaled for identification purposes by fixing the coefficient of one of the causal indicators [70]. Childhood environmental adversity was scaled by setting the path from "parental education" to 1. Fourth, the reflective latent variable reproduction-maintenance trade-off and the reflective latent variable collective action were scaled by fixing their variance to 1. Fifth, the correlation between the residual errors of "age at first birth" and "number of children" was included in the model. The reason for this is that, all else being equal, the sooner you have children, the more children you can have. This correlation is not captured by the latent variable reproduction-maintenance trade-off and is therefore included separately; these two variables indeed covary mechanically [70].

Hooper et al. [97] recommend determining model fit by using $\chi^2$, Comparative Fit Index (CFI), Root Mean Square Error of Approximation (RMSEA), and Standardized Root Mean Square Residual (SRMR). A good fit is assumed if the indices are close to the following values: $\chi^2$ p-value $> 0.05$, CFI $> 0.95$, RMSEA p-value $< 0.07$ and SRMR $< 0.08$. It is important to note however, that the large sample size of our study prevents us from interpreting $\chi^2$ as evidence for a discrepancy between the sample and the model-implied covariance matrix. The chi-square statistic is indeed known to be particularly sensitive to sample size, which can lead models fitted on large samples to be systematically rejected [98]. Furthermore, the $\chi^2$ assumes normality [99] and the deviation from normality characterizing some of our variables might result in the rejection of the model even when the model is properly specified. Therefore, only the scaled versions of the CFI, RMSEA and SRMR are reported, which eliminates the issues of sample size dependency and non-normality of continuous and categorical data [100].

**2.1.4 Covariates.** Given that cooperation is affected by age, we included age as an auxiliary variable to control for its effect on the collective action variables. Freund & Blanchard-Fields [101], for example, found that older adults report valuing contributions to the public good more positively and are more likely to behave altruistically than younger adults. Furthermore, age is also used as an auxiliary variable to control for its effect on the "subjective health status" variable and on the "number of children" variable. The reason for this is that, all else being equal, the older you get the poorer your health is likely to be and the more children you are likely to have.

Childhood environmental adversity and current environmental adversity are moderately correlated [51]. Therefore, we ran an additional analysis which was not pre-registered where we controlled for the effect of current adversity on the reproduction-maintenance trade-off and collective action variables. Current adversity was modeled using the respondent's current income level (scale 1–10: the higher the score the lower the income level).

**2.1.5 Analyses.** We first carried out our analyses on the discovery sample; we then preregistered our analysis plan on the Open Science Framework (https://osf.io/642hd); finally, we applied the pre-registered analyses on the replication sample. The results of the replication sample are presented below. The results of the discovery sample are detailed in S1 Text; the significance and direction of the key associations are consistent with the replication sample. Although, two additional indicators for childhood environmental adversity reach significance: "parents had problems making ends meet" and "death of mother".

All statistical analyses were carried out in R 3.4.4 (https://www.r-project.org/) with R Studio 1.1.456. The structural equation model was fitted using the R package *lavaan* [102]. The WLSMV estimator was used for its robustness to departures from normality [102].

When data was missing, we conducted listwise deletions. Although our variables of interest showed overall low percentages of missing responses (ranging from 0 to 13%), an additional analysis was done on the replication sample with imputed data. The results of this analysis are largely consistent with the results obtained using listwise deletions (see S2 Text). The main difference is that childhood environmental adversity is significantly–albeit weakly—associated with increased adult involvement in collective action–a result that we will discuss in more detail below.

For the mediation analyses the bootstrap method developed by Preacher & Hayes [103] was used, which is recommended by MacKinnon et al. [104]. This is a non-parametric resampling test. The main feature of this test is that it does not rely on the assumption of normality. Bootstrapping estimates the upper limit and the lower limit of the confidence intervals of an indirect effect. We computed bootstrapped 95% confidence intervals (1000 bootstrap samples). The direct effect, indirect effect and total effect can be found in S3 Text.

## 2.2 Results

### 2.2.1 Correlation matrix and descriptive statistics.
Descriptive statistics and the correlation matrix for the variables included in the structural equation model can be found in Table 1.

### 2.2.2 Model fit.
The scaled CFI value (0.900), the scaled RMSEA value (0.040) and the scaled SRMR value (0.018) are fairly consistent with a close-fitting model. Therefore, the approximate fit indices reveal no strong misspecification for this model.

### 2.2.3 Measurement model.
The standardized regression weights can be found in Fig 1 (the full results can be found in S3 Table). Childhood environmental adversity is driven by two main contributors: "parents had problems replacing things" (UnStd $c$ = 2.37 (0.26), $z$ = 9.01, $p < 0.001$, Std $c$ = 0.69) and "death of the father before age 16" (UnStd $c$ = 1.65 (0.70), $z$ = 2.37, $p < 0.05$, Std $c$ = 0.09). "Death of the mother before age 16" and "parents had problems makings ends meet" did not load significantly on childhood environmental adversity.

"Subjective health status" (UnStd $c$ = -0.39 (0.02), $z$ = -25.97, $p < 0.001$, Std $c$ = -0.43) and "age at first birth" (UnStd $c$ = -1.60 (0.07), $z$ = -24.63, $p < 0.001$, Std $c$ = -0.33) loaded

**Table 1. European Values Study descriptive statistics and correlations.**

|  | (1) | (2) | (3) | (4) | (5) | (6) | (7) | (8) | (9) | (10) | (11) |
|---|---|---|---|---|---|---|---|---|---|---|---|
| (1) Parental education | - | | | | | | | | | | |
| (2) Parents problem making ends meet | 0.28* | - | | | | | | | | | |
| (3) Parents problem replacing things | 0.24* | 0.71* | - | | | | | | | | |
| (4) Death of father | 0.07* | 0.11* | 0.10* | - | | | | | | | |
| (5) Death of mother | 0.02* | 0.01 | 0.01 | 0.03* | - | | | | | | |
| (6) Health | -0.16* | -0.16* | -0.18* | -0.06* | -0.02* | - | | | | | |
| (7) Age 1st birth | -0.06* | -0.08* | -0.08* | -0.02 | -0.01 | 0.11* | - | | | | |
| (8) Number of children | 0.19* | 0.10* | 0.07* | 0.02 | 0.03* | -0.08* | -0.18* | - | | | |
| (9) Volunteering | -0.03* | -0.02* | -0.03* | -0.01 | 0,00 | 0.1* | 0.05* | 0.04* | - | | |
| (10) Political action | -0.1* | -0.07* | -0.11* | -0.02* | 0,00 | 0.15* | 0.13* | -0.01 | 0.12* | - | |
| (11) Age | 0.31* | 0.16* | 0.13* | 0.11* | 0.05* | -0.33* | 0.1* | 0.24* | 0,00 | -0.01 | - |
| Mean | 5.49 | 2.40 | 2.34 | 0.05 | 0.02 | 3.56 | 25.18 | 2.21 | 0.38 | 0.61 | 51.96 |
| SD | 1.99 | 1.17 | 1.17 | 0.22 | 0.13 | 0.96 | 5.18 | 1.12 | 1.21 | 0.96 | 15.00 |
| Range | 1–8 | 1–4 | 1–4 | 0–1 | 0–1 | 1–5 | 12–83 | 1–13 | 0–15 | 0–5 | 18–108 |

* $p < 0.05$.

significantly on the reproduction-maintenance trade-off. The pattern of covariation follows our predictions: poorer reported health and a younger age at first child's birth. "Number of children" did not load significantly on the reproduction-maintenance trade-off. However, inspection of the estimated covariance shows that "number of children" is not independent but covaries with "age at first birth" in the expected way (UnStd c = -1.15 (0.04), z = -31.63, p < 0.001, Std c = -0.22), specifically a higher age at first child's birth correlates with lower number of children, an observation that is in line with existing findings [70]. Hence, the reproduction-maintenance trade-off is fairly consistent with prior studies [69,70].

"Volunteering" (UnStd c = 0.34 (0.02), z = 17.12, p < 0.001, Std c = 0.52) and "political action" (UnStd c = 0.40 (0.03), z = 14.90, p < 0.001, Std c = 0.61) loaded significantly on the collective action latent variable, whose greater values indicate higher investments in both volunteering and political activities.

**2.2.4 Structural model.** Fig 1 shows that the direct effect of childhood environment adversity is significantly–albeit weakly–associated with decreased adult involvement in collective action (UnStd c = -0.03 (0.01), z = -3.50, p < 0.001, Std c = -0.08). This implies that an adverse childhood environment is associated with less investment in collective actions later in life, relatively independently of variations in individuals' reproduction-maintenance trade-off. Furthermore, an adverse childhood environment is associated with variations in individuals' reproduction-maintenance-trade-off (UnStd c = 0.08 (0.01), z = 10.46, p < 0.001, Std c = 0.29), specifically an increased investment in reproduction and a decreased investment in somatic maintenance. The reproduction-maintenance trade-off is itself associated with lower adult involvement in collective action (UnStd c = -1.07 (0.10), z = -10.87, p < 0.001, Std c = -0.73). In line with our first hypothesis, a significant part of the effect of childhood environmental adversity on adult involvement in collective action is mediated by the reproduction-maintenance trade-off (indirect effect: UnStd c = -0.0027 (0.0004), bootstrapped ci lower = -0.0035, bootstrapped ci upper = -0.0020, z = -6.92, p < 0.001).

The above mentioned effects remain stable after the inclusion of the respondent's current adversity as an additional covariate (scaled CFI = 0.897; scaled RMSEA = 0.042; scaled SRMR = 0.017), and the magnitude of key parameters stayed significant and in the expected direction (the full results can be found in S4 Table).

Given that the mediation effect is strong, the direct effect of childhood environmental adversity on collective action is to be interpreted with caution. Our analyses show that it is indeed a small and unstable effect, which reverses in the dataset that includes imputed data (UnStd c = 0.04 (0.02), z = 2.23, p < 0.05, Std c = 0.13). To confirm that this coefficient instability can be attributed to the presence of a mediator capturing a large part of the effect, we ran an additional model on the replication sample with imputed data, but without the mediating variable (i.e., reproduction-maintenance trade-off). This analysis revealed that adverse childhood environments are indeed associated with decreased collective action (UnStd c = -0.07 (0.01), z = -7.09, p < 0.001, Std c = -0.21).

The other regression coefficients are larger and more stable across analyses and datasets (i.e., the dataset implementing listwise deletions and the imputed dataset). This is the case for the regression between childhood environmental adversity and the reproduction-maintenance trade-off (listwise deletions UnStd c = 0.08 (0.01), z = 10.46, p < 0.001, Std c = 0.29; imputed data UnStd c = 0.07 (0.01), z = 6.92, p < 0.001, Std c = 0.39), for the regression between the reproduction-maintenance trade-off and collective action (listwise deletions UnStd c = -1.07 (0.10), z = -10.87, p < 0.001, Std c = -0.73; imputed data UnStd c = -1.50 (0.34), z = -4.42, p < 0.001, Std c = -0.88), as well as for the mediation effect (listwise deletions indirect effect: UnStd c = -0.0027 (0.0004), bootstrapped ci lower = -0.0035, bootstrapped ci upper = -0.0020, z = -6.92, p < 0.001; imputed data indirect effect: UnStd c = -0.0023 (0.0004), bootstrapped ci

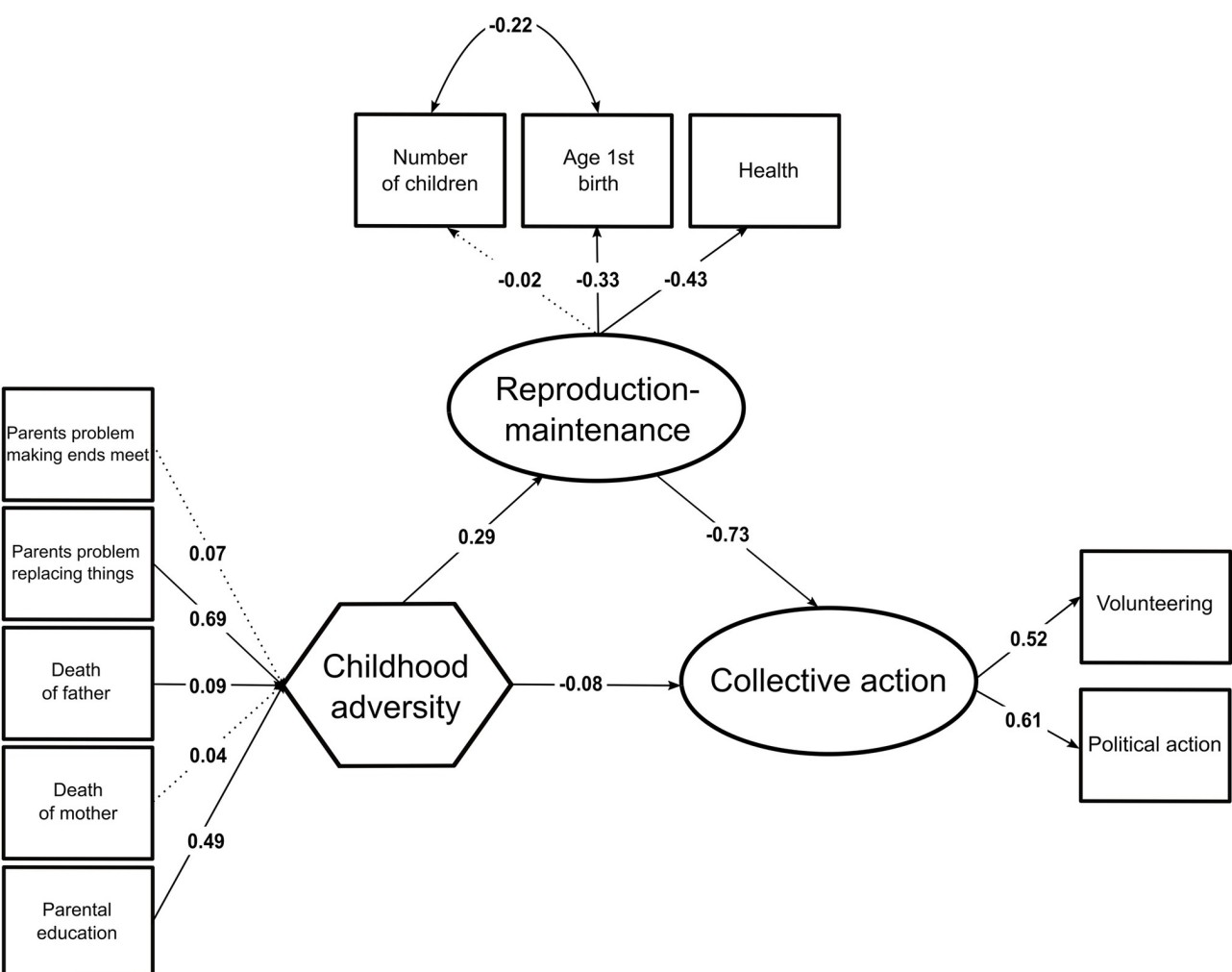

**Fig 1. European Values Study standardized parameter values estimated by the structural equation model.** Significant paths at the 5% level are represented with a continuous arrow.

lower = -0.0030, bootstrapped ci upper = -0.0016, $z$ = -6.64, $p < 0.001$). Overall, these results converge on the idea that the relationship between childhood environmental adversity and collective action is largely conditional on the reproduction-maintenance trade-off.

Finally, the range for "age at first child's birth" is from 12 to 83 years old. This range is theoretically possible but unlikely (especially for women). We suspect that some of these data points are in fact entry errors. However, we kept the full sample for two reasons: first, it is impossible to tease apart entry errors from true late first births and second, very few individuals reported to have had their first birth at a later age (only 14 women out of 17.087). To ensure that the inclusion of these 14 participants did not alter our results we ran the model again without these participants and found the same overall pattern.

**2.2.5 Conclusions.** Our prediction was that people who grew up in adverse environments invest less in collective actions later in life and that this effect is mediated by modulations of the reproduction-maintenance trade-off. Data from the European Values Study partly confirm this hypothesis and show that the association between childhood environmental adversity and

adult involvement in collective action is partly mediated by variations in individuals' repro-duction-maintenance-trade-off.

# 3. Study 2: Current adversity and collective action in the World Values Survey

We then had the objective of replicating the pattern of associations found in study 1 on an independent sample extracted from the World Values Survey dataset [105]. The World Values Survey has relied on social scientists since 1981 to collect data from large representative samples of the populations from almost 100 European and non-European countries. In this study, we focused on the fourth wave (1999–2004) for two reasons: first, it includes non-European countries; second, unlike the other waves, the fourth wave includes the same collective action variables as the ones included in the European Values Study sample analyzed in study 1.

It is worth acknowledging that using the World Values Survey fourth wave also comes with several limitations. First, unlike the European Values Study fourth wave, the World Values Survey fourth wave (and the other waves as well) does not contain any data about the respondents' age at first reproduction. Therefore, the reproduction-maintenance trade-off can only be modeled on two indicators: the respondents' number of children and their self-reported health status. Nevertheless, as we point in section 2.1.2 of the present article, the existing litera-ture provides substantial evidence that the number of children and the self-reported health sta-tus can be considered as fairly good proxies for the reproduction-maintenance trade-off. The second limitation is that the World Values Survey fourth wave contains no data on the respon-dents' childhood environment. We therefore focus on respondents' current income as an indi-cator of the level of adversity they are currently exposed to. This choice is primarily motivated by a number of studies showing that income correlates with almost all forms of morbidity and mortality [2–4], such that economically deprived people also experience greater exposure to disability and death. The second motivation comes from the fact that economic status in adult-hood generally correlates with economic status in childhood [106,107].

## 3.1 Methods

**3.1.1 Respondents.** 59.030 respondents living in 40 different countries are included in wave 4 of the World Values Survey. However, some respondents could not be included in our analyses. First, one of the variables (i.e., number of children) used to model the reproduction-maintenance-trade-off is only informing for respondents with children. The reason is that peo-ple who do not have children now fall into three categories: (1) people who do not have chil-dren yet but will do so in the future, (2) people who do not have children yet and will choose not to do so in the future and (3) people who do not have children yet but would like to and can't. Since we have no way of distinguishing these three groups, we focused on the people who do have children, which left us with the 41.799 respondents who report having children (this also makes it consistent with the first study, where we also removed respondents without children). Second, we removed countries for which the collective action variables had not been filled out, which reduced the dataset to 25.670 respondents living in 27 different countries. We randomly split this sample into a discovery subsample (12.962 respondents) and a replication subsample (12.708 respondents). In both subsamples, we excluded respondents with too much missing data (the criterion was set to more than two standard deviations from a respondent's respective subsample mean) which led to 12.083 respondents for the discovery sample and 11.888 respondents for the replication sample. The final discovery sample after listwise dele-tions included 10.829 respondents (females: N = 5.633; mean age = 45 ± 15 sd; mean number of respondents per country = 417 ± 150 sd) and the final replication sample after listwise

deletions included 10.617 respondents (females: N = 5.589; mean age = 45 ± 14 sd; mean number of respondents per country = 408 ± 157 sd). The final replication sample after listwise deletions consists of 26 countries (descriptive statistics for each country are detailed in S5 Table).

**3.1.2 Variables.**   We selected variables tapping three constructs: current adversity, the reproduction-maintenance-trade-off and collective action. Respondents' current adversity was represented by their current income level (scale 1–10: the higher the score the lower the income level) [2–4,51].

The reproduction-maintenance-trade-off was modeled by constructing a reflective latent variable based on "number of children" and "subjective health status" (scale 1–5: the higher the score the better the respondents' reported health).

The reflective latent construct "collective action" was estimated from the very same indicators as in study 1. The only difference is that scores in "volunteering" and "political action" were calculated on the basis of 14 and 4 items respectively (instead of 15 and 5 items in study 1, respectively) (items are detailed in S6 Table).

**3.1.3 Model specification and fit.**   The model's structure was the same as the one specified in study 1, with two notable differences. First, childhood environmental adversity was replaced by current adversity. Second, "age at first birth" is no longer an indicator of the latent variable reproduction-maintenance-trade-off. Age was again used as an auxiliary variable to control for its effect on the collective action variables and "subjective health status" variable and "number of children" variable. Model fit was determined in the same manner as in study 1.

**3.1.4 Analyses.**   We first carried out our analyses on the discovery sample; we then preregistered our analysis plan on the Open Science Framework (https://osf.io/37bs2/); finally, we applied the pre-registered analyses on the replication sample. The results of the replication sample are presented below (the results of the discovery sample are detailed in S4 Text; the significance and direction of the key associations are consistent with the replication sample).

When data was missing, we conducted listwise deletions (as in study 1). Although our variables of interest showed overall low percentages of missing responses (ranging from 0 to 13%), an additional analysis was done on the replication sample with imputed data. The results of this analysis are consistent with the results obtained using listwise deletions (see S5 Text).

All statistical analyses and the method for mediation analyses were similar to those reported in study 1. The direct effect, indirect effect and total effect can be found in S6 Text.

## 3.2 Results

**3.2.1 Correlation matrix and descriptive statistics.**   Descriptive statistics and the correlation matrix for the variables included in the structural equation model can be found in Table 2.

**3.2.2 Model fit.**   The scaled CFI value (0.960), the scaled RMSEA value (0.047) and the scaled SRMR value (0.018) are consistent with a close-fitting model. Therefore, the approximate fit indices reveal no strong misspecification for this model.

**3.2.3 Measurement model.**   The standardized regression weights can be found in Fig 2 (the full results can be found in S7 Table). "Subjective health status" (UnStd c = -0.15 (0.03), $z$ = -4.38, $p < 0.001$, Std c = -0.24) and "number of children" (UnStd c = 0.22 (0.05), $z$ = 4.39, $p < 0.001$, Std c = 0.17) loaded significantly on the reproduction-maintenance-trade-off. As in study 1, the pattern of covariation follows our predictions: poorer reported health and higher number of children. Hence, the reproduction-maintenance-trade-off is consistent with prior studies [69,70].

"Volunteering" (UnStd c = 0.20 (0.04), $z$ = 5.69, $p < 0.001$, Std c = 0.12) and "political action" (UnStd c = 0.63 (0.16), $z$ = 4.01, $p < 0.001$, Std c = 0.75) loaded significantly on the

**Table 2. World Values Survey descriptive statistics and correlations.**

|  | (1) | (2) | (3) | (4) | (5) | (6) |
|---|---|---|---|---|---|---|
| (1) Current adversity | - |  |  |  |  |  |
| (2) Health | -0.18* | - |  |  |  |  |
| (3) Number of children | 0.13* | -0.09* | - |  |  |  |
| (4) Volunteering | -0.04* | 0.03* | 0.06* | - |  |  |
| (5) Political action | -0.13* | 0.06* | -0.05* | 0.09* | - |  |
| (6) Age | 0.03 | -0.21* | 0.24* | -0.05* | 0.06* | - |
| Mean | 6.56 | 3.78 | 2.93 | 1.03 | 0.57 | 44.53 |
| SD | 2.35 | 0.87 | 1.78 | 1.88 | 0.94 | 14.45 |
| Range | 1–10 | 1–5 | 1–8 | 0–14 | 0–4 | 16–98 |

* $p < 0.05$.

collective action latent variable, whose greater values indicate higher investments in political activities and, to a lesser extent, higher investments in volunteering activities.

**3.2.4 Structural model.** Fig 2 shows that the direct effect of current adversity on adult involvement in collective action is not significant (UnStd c = 0.12 (0.11), $z$ = 1.12, $p$ = 0.26, Std

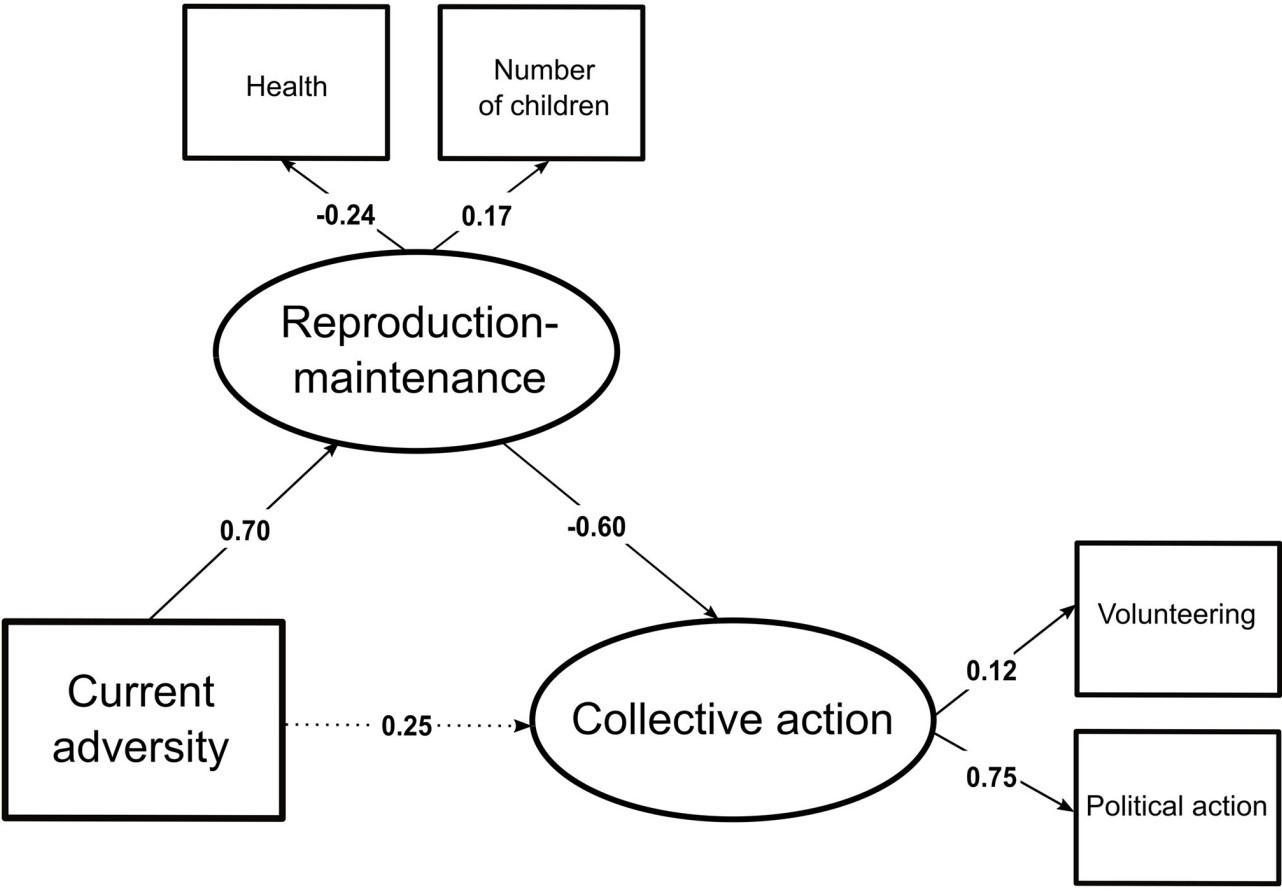

**Fig 2. World Values Survey standardized parameter values estimated by the structural equation model.** Significant paths at the 5% level are represented with a continuous arrow.

c = 0.25). Furthermore, more current adversity is associated with variations in individuals' reproduction-maintenance trade-off (UnStd c = 0.42 (0.10), $z$ = 4.36, $p < 0.001$, Std c = 0.70), specifically an increased investment in reproduction and a decreased investment in somatic maintenance. The reproduction-maintenance-trade-off is itself associated with lower adult involvement in collective action (UnStd c = -0.48 (0.19), $z$ = -2.49, $p < 0.05$, Std c = -0.60). In line with our second hypothesis, the effect of current adversity on adult involvement in collective action is mediated by the reproduction-maintenance trade-off (indirect effect: UnStd c = -0.003 (0.001), bootstrapped ci lower = -0.004, bootstrapped ci upper = -0.001, $z$ = -2.67, $p < 0.01$).

Despite the fact that the association pattern reported above is identical to the one found with the European Values Study (EVS) data, one should note that the effects are smaller (by comparing the z-values). This is the case for the regression between environmental adversity and collective action (EVS UnStd c = -0.03 (0.01), $z$ = -3.50, $p < 0.001$, Std c = -0.08; WVS UnStd c = 0.12 (0.11), $z$ = 1.12, $p$ = 0.26, Std c = 0.25), between environmental adversity and the reproduction-maintenance trade-off (EVS UnStd c = 0.08 (0.01), $z$ = 10.46, $p < 0.001$, Std c = 0.29; WVS UnStd c = 0.42 (0.10), $z$ = 4.36, $p < 0.001$, Std c = 0.70), between the reproduction-maintenance trade-off and collective action (EVS UnStd c = -1.07 (0.10), $z$ = -10.87, $p < 0.001$, Std c = -0.73; WVS UnStd c = -0.48 (0.19), $z$ = -2.49, $p < 0.05$, Std c = -0.60) as well as for the mediation effect (EVS indirect effect: UnStd c = -0.0027 (0.0004), bootstrapped ci lower = -0.0035, bootstrapped ci upper = -0.0020, $z$ = -6.92, $p < 0.001$; WVS indirect effect: UnStd c = -0.003 (0.001), bootstrapped ci lower = -0.004, bootstrapped ci upper = -0.001, $z$ = -2.67, $p < 0.01$).

**3.2.5 Conclusions.** Our prediction was that people who currently live in adverse environments invest less in collective action and that this effect is mediated by modulations of the reproduction-maintenance trade-off. Data from the World Values Survey confirm this hypothesis and show that the association between current adversity and adult involvement in collective action is mediated by variations in individuals' reproduction-maintenance trade-off.

## 4. Discussion

The present study replicates existing research linking adverse environments in childhood and adulthood with decreased investment in adult cooperation [33,34,36–38], and finds that this association is mediated by variations in individuals' reproduction-maintenance-trade-off. Furthermore, it adds to the consistent pattern of results (i.e., a relationship between adverse environments and decreased cooperation) found by existing studies that use self-reported questionnaires to measure cooperation [33,34].

Even though the findings reported above are consistent with the existing literature, it is important that we highlight several limitations. First, we used people's involvement in collective actions as a proxy for cooperation but cooperation obviously encompasses more than collective action behaviors. The reason we chose these items is that, unlike self-reported attitudes about cooperative morals often used as proxies for cooperation in other survey studies [90], these were the only variables that met our a priori defined criteria (actual behaviors involving concrete costs and delayed benefits for the agent and have benefits for others that had previously been successful in predicting real-life cooperation) in the European Values Study and the World Values Survey. However, people might cooperate in different ways than in large-scale collective action. For example, they might actively cooperate within more restricted social networks centered around kinship [108]. Thus, it remains an open question whether people cooperate less under adverse conditions in the broadest sense or simply invest less in collective actions.

Second, our goal was to investigate if part of the variability in people's involvement in collective action was mediated by differences in individuals' reproduction-maintenance-trade-off. Although the reproduction-maintenance-trade-off is well captured in the World Values Survey model, it is less well captured in the European Values Study model. Specifically "number of children" is not significantly correlated with the reproduction-maintenance trade-off. This may reflect the fact that the reproduction-maintenance trade-off indicators are influenced by multiple causes beyond the reproduction-maintenance trade-off itself, including cultural factors like contraceptives. Such cultural factors might account for the fact that "number of children" is not significantly correlated with respondents' reproduction-maintenance trade-off in the European Values Study [109].

Third, the large sample sizes used in our studies raise the possibility that the statistically significant associations reported are primarily due to high statistical power. Therefore, the reporting of p-values is less informative. Standardized regression coefficients can be used as an effect size index [110]. Standardized regression coefficients refer to how many standard deviations the dependent variable will change per a standard deviation increase in the independent variable, holding all other variables constant. Standardized regression coefficients can be interpreted similar to correlations, where <0.2 is considered a weak effect, between 0.2 and 0.5 as a moderate effect and >0.5 as a strong effect [111]. Looking at our standardized path coefficients; the effect between adversity and collective action is weak (EVS Std c = -0.08; WVS not significant), the effect between adversity and the reproduction maintenance trade-off is moderate for the European Values Study and strong for the World Values Survey (EVS Std c = 0.29; WVS Std c = 0.70) and the effect between the reproduction maintenance trade-off and collective action is strong (EVS Std c = -0.73; WVS Std c = -0.60).

Fourth, it is important to note that life-history theory has been hotly debated for theoretical, empirical and methodological reasons in recent years [112–116]. Of particular relevance to this paper is the debate questioning whether life-history theory can be applied to explain individual differences in humans. Life-history theory was initially developed to explain differences between species [117] and later applied to individual differences in humans by invoking genetic factors, early environment conditions, or phenotypic plasticity [1,73,77,118], the latter two concepts being central to the present paper. One criticism laid out by Zietsch & Sidari [116] is that the processes that create between species trait covariation (i.e., Darwinian evolution) and within species trait covariation (e.g., developmental plasticity) are fundamentally different. In response to this, Del Giudice [119] has argued that the two processes can in fact be functionally related. He proposed the term "ecological gambit", where trade-offs in the context of life-history strategies form the basis for the functional link between population and individual differences, as is the case in our analyses focusing on the reproduction-maintenance trade-off. In addition, there is evidence that phenotypic plasticity is the dominant source of inter-individual and inter-population variation in reproductive traits and associated health costs [66,120].

More generally, the correlational nature of our data hinders the possibility of producing inferences about the causal role of adverse environments on cooperation. For example, it is possible that the relationship between environmental adversity and cooperation is reversed: adverse environments might be associated with lower cooperation because individuals who cooperate less are less likely to do well in life and therefore more likely to end up living in a adverse environment. This alternative is plausible albeit a little harder to reconcile with the fact that we also found an association between an adverse childhood environment and adult cooperation. Another possibility is that there is an unknown variable that influences both environmental adversity and cooperation. Finally, the relationship between environmental adversity

and cooperation might not be monotonic. For example, Lazarus [121] has proposed that cooperation may be an inverted-U function of environmental adversity.

One way to demonstrate the causal effect of environmental adversity on cooperation would be to experimentally manipulate acute stress [13], for example by administering hydrocortisone (a synthetic form of cortisol). Using this methodology, Riis-Vestergaard et al. [122] found that administering hydrocortisone increases temporal discounting over the short term. Another possibility to induce stress is to expose people to mortality cues [51], e.g. in the form of news articles stating that in the local area there is an increase in violence. However, it is important to note that there are large differences between the effects of acute stress and chronic stress [123]. For example, acute stress has been found to increase physical arousal, learning, motivated behavior and sensation seeking. By contrast, chronic stress has been found to impair attentional control, behavioral flexibility and it can promote anxiety, depression and learned helplessness.

Another option to identify causal relationships is to analyze longitudinal data involving exogenous shocks to the individual's environment (e.g., sudden increase of income or in contrast famine, war, etc.). There have been a handful of longitudinal studies testing the effect of stress during childhood and social behavior in adulthood using exogenous shocks of positive or negative income on different behavioral outcomes. These studies have shown that higher levels of income tend to make individuals more 'agreeable' and more trustful [27,124] while higher levels of violence during childhood make people more antisocial and more violent [125,126]. One of the most noteworthy pieces of research in this area is a quasi-experimental study led by Akee et al. [124]. In this study, American Indian children were exposed to a positive income shock following the opening of a casino that decided to allocate part of its revenue to Cherokee households. Particularly interesting is that the increase in income led to an increase in agreeableness, which correlates with cooperation and unselfishness [127,128]. Hence, at least part of the effects of childhood environmental adversity on cooperation might capture long-term plastic responses to the environment.

It should be noted that our results are compatible with a range of mechanisms. The standard mechanism put forward in the literature (and in our introduction) is that cooperation is adaptively reduced in adverse environments because individuals have a shorter life expectancy and should therefore be more short-term oriented [47]. However, the idea that people focus more on the present because they have a higher probability of dying at age 60 instead of age 80 is problematic. Why should this affect their interest in cooperation today? The fact that someone is alive at age 60 or age 80 indeed has little impact on his or her probability of being alive tomorrow [129,130]. A recent theoretical model demonstrates that waiting costs (i.e., the cost paid by an individual even when the benefits are guaranteed) are another mechanism by which adverse environments can be associated with higher discount rates [131], and therefore decreased cooperation. When an individual has to choose between obtaining a resource immediately or a larger resource later she must take into account all the benefits she could have earned, during the delay, by wisely using this resource. Resources can be transformed into capital which allows individuals to better exploit their environment. When an individual chooses a delayed resource over an immediate one, she therefore renounces all the benefits this supplementary capital would have generated during that delay. These forgone benefits are independent of mortality or other collection risk. They are paid even if the individual is certain to obtain the resource eventually. They constitute the waiting cost per se. The more an individual can gain by investing new resources wisely into capital, the higher the waiting cost. The models shows that individuals who have already accumulated a lot of personal capital, and are high up on their pyramid of needs, should be relatively patient because each supplementary unit of capital makes a small difference to them anyway, whereas those who have accumulated little

and are still at the bottom of their pyramid of needs should have a short time horizon as each additional resource can make a big difference to them.

Another mechanism by which environmental adversity can have an effect on cooperation is risk management. For example, Amir et al. [39] put forward an uncertainty management perspective, where adverse environments are associated with risk averse, present-oriented and more cooperative preferences. Although, they do not specifically test whether risk is a mediator between environmental adversity and cooperation. Even in the uncertainty management perspective proposed by Amir et al. [39], adverse environments are associated with a present-orientation, which is in line with the framework of our study. Amir et al. [39] see cooperation as a form of risk pooling which would reduce uncertainty. Therefore, in adverse and uncertain environments, this would be associated with increased cooperation. We fully agree that in adverse environments people will want to reduce uncertainty, but we are not entirely convinced that the primary means to achieve this is increased cooperation. One possibility is that this might be the case in specific contexts, for example, where population size is low and where the likelihood of interacting with a third party outside the kinship community is small. However, the pattern might be reversed in places where the population size is larger and where the likelihood of interacting with non-kin increases, because the likelihood of being cheated also increases as a result [132–134], and this risk might further increase in adverse conditions. In these environments, trusting other people less and cooperating less can be viewed as a means to reduce uncertainty, a view which is in contradiction with Amir et al.'s findings [39], but which fits a large set of work in the social sciences [81,83,135] as well as our own results. We think that risk management strategies in the context of social behaviors is an interesting topic for further research.

Whether cooperation is decreased or increased in adverse environments is still debated. The present study contributes to this debate by reporting two pre-registered studies applying structural equation models to test this relationship on large-scale datasets (the European Values Study and the World Values Survey). The results show that adverse environments in childhood and adulthood are associated with decreased investment in adult cooperation. Furthermore, the channel through which this relationship arises is also unclear. We proposed that this relationship is mediated by a present bias at the psychological level, which is embodied in the reproduction-maintenance trade-off at the biological level. We indeed found that the relationship between environmental adversity and cooperation is mediated by individuals' reproduction-maintenance trade-off.

## Supporting information

**S1 Table. European Values Study countries descriptive statistics.** Values are unstandardized coefficients with standard deviations in brackets.
(XLSX)

**S2 Table. European Values Study collective action items.**
(DOCX)

**S3 Table. European Values Study listwise deletions—Full results.**
(XLSX)

**S4 Table. European Values Study listwise deletions—Controlled for current adversity.**
(XLSX)

**S5 Table. World Values Study countries descriptive statistics.** Values are unstandardized coefficients with standard deviations in brackets.
(XLSX)

**S6 Table. World Values Survey collective action items.**
(DOCX)

**S7 Table. World Values Survey listwise deletions—Full results.**
(XLSX)

**S1 Text. European Values Study discovery sample—Results.**
(DOCX)

**S2 Text. European Values Study imputed data—Results.**
(DOCX)

**S3 Text. European Values Study listwise deletions—Mediation effects.**
(DOCX)

**S4 Text. World Values Survey discovery sample—Results.**
(DOCX)

**S5 Text. World Values Survey imputed data—Results.**
(DOCX)

**S6 Text. World Values Survey listwise deletions—Mediation effects.**
(DOCX)

## Author Contributions

**Conceptualization:** N. Lettinga, P. O. Jacquet, C. Chevallier.

**Data curation:** N. Lettinga, P. O. Jacquet.

**Formal analysis:** N. Lettinga, P. O. Jacquet.

**Funding acquisition:** C. Chevallier.

**Methodology:** N. Lettinga, P. O. Jacquet.

**Project administration:** N. Lettinga.

**Supervision:** N. Baumand, C. Chevallier.

**Validation:** P. O. Jacquet, J-B. André, N. Baumand, C. Chevallier.

**Visualization:** N. Lettinga.

**Writing – original draft:** N. Lettinga.

**Writing – review & editing:** P. O. Jacquet, J-B. André, N. Baumand, C. Chevallier.

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
