## [Decision Letter · Decision Letter 0]

11 Mar 2020

PONE-D-20-00738

Environmental harshness is associated with lower investment in collective actions

PLOS ONE

Dear Mr Lettinga,

Thank you for submitting your manuscript to PLOS ONE. After careful consideration, we feel that it has merit but does not fully meet PLOS ONE’s publication criteria as it currently stands. Therefore, we invite you to submit a revised version of the manuscript that addresses the points raised during the review process.

We would appreciate receiving your revised manuscript by Apr 25 2020 11:59PM. To enhance the reproducibility of your results, we recommend that if applicable you deposit your laboratory protocols in protocols.io, where a protocol can be assigned its own identifier (DOI) such that it can be cited independently in the future. For instructions see: http://journals.plos.org/plosone/s/submission-guidelines#loc-laboratory-protocols

We look forward to receiving your revised manuscript.

Kind regards,

Valerio Capraro

Academic Editor

PLOS ONE

Journal Requirements:

Additional Editor Comments (if provided):

The reviewers like the paper but suggest several improvements. Therefore, I would like to invite you to revise your manuscript according to their comments. Needless to say that all comments must be addressed.

I am looking forward for the revision.

Reviewers' comments:

Reviewer's Responses to Questions

**Comments to the Author**

1. Is the manuscript technically sound, and do the data support the conclusions?

Reviewer #1: Yes

Reviewer #2: Yes

2. Has the statistical analysis been performed appropriately and rigorously? 

Reviewer #1: I Don't Know

Reviewer #2: Yes

3. Have the authors made all data underlying the findings in their manuscript fully available?

Reviewer #1: Yes

Reviewer #2: Yes

4. Is the manuscript presented in an intelligible fashion and written in standard English?

Reviewer #1: Yes

Reviewer #2: Yes

5. Review Comments to the Author

Reviewer #1: Environmental harshness is associated with lower investment in collective actions

Lettinga, N., Jacquet, P.O., André, J-B., Baumard, N., Chevallier, C.

Reviewer: John Lazarus

Limitations of review. I am not expert in structural equation modelling and I do not have experience working with either of the data sets used here.

General comments

1. This is a valuable study in a currently lively and somewhat controversial area of research. It is particularly useful in measuring associations of prosociality with both environmental harshness and measures predicted by life history strategy (LHS) theory.

2. The paper tests the hypotheses that prosociality is reduced in adults both in a harsher current environment and when raised in a harsher environment during childhood. The paper is clearly written, the Introduction covers the relevant literature well and the Discussion section is commendable for the way it considers limitations of the study; problems of causal inference from correlational data; and alternative evolutionary explanations for the findings.

3. Throughout the paper the authors describe various behavioural measures examined by others, and by themselves in this paper, as ‘cooperation’. In evolutionary biology cooperation is defined as a social action resulting in a benefit to both (all) parties in the interaction. The behaviours analysed here are volunteering and political action, behaviours that would be defined in evolutionary biology as ‘altruism’; that is, actions resulting in a benefit to the recipient(s) and a cost to the self, and the authors agree that these two behaviours do have this cost-benefit structure (lines 187, 189-90 and 475 [later numerical references are also to line numbers]). Similarly some behavioural measures in previous studies cited here as cooperative (e.g. reference 30) are actually altruistic (e.g. volunteering and charitable giving).

Since this paper is testing hypotheses within the discipline of evolutionary human studies it is important that the terminology is correct. This is far from being a dry terminological matter; rather it is central to the hypotheses tested here, as I shall go on to argue. The authors are testing the idea that in harsh

2

environments there is a ‘need to favour immediate over long-term benefits’ (54) and they state that ‘cooperation is a future-oriented strategy: in the short term, it is more advantageous to reap immediate, smaller, but more certain benefits by cooperating less’ (62-64).

I have a number of comments related to this point: • (a) The behavioural measures employed here – volunteering and political action – are altruistic not cooperative as defined above. Consequently they are suitable for testing the hypothesis that they will be less evident in harsh environments, but this is because they are immediately costly, and not, as argued in the paper, because benefits from them are delayed. • (b) I do not agree with the claim quoted above (62-64, also 19-20) that cooperation is necessarily a ‘future-oriented strategy’. It is futureoriented in the case of reciprocal altruism where, after an exchange of altruistic acts between two individuals, both have gained a net benefit (but is only future-oriented if the focal individual is the first altruist). (Volunteering and political action, in themselves, do not fall into this category – rather they are altruistic – although one can imagine scenarios in which they are followed by a reciprocal benefit.) But where the cooperation involves joint simultaneous activity there is an immediate benefit to both parties (compared to not cooperating). • (c) In consequence of this last point, and following the arguments of the authors, simultaneous cooperation, as defined here, would be predicted to be more common under harsher conditions since it provides an immediate benefit in comparison with the decision to refrain from cooperation. • (d) In using a ‘proxy’ measure for cooperation (e.g. 95-6) it is necessary that the proxy satisfies the definition of cooperation given above. I do not mean here the distinction between behavioural and attitudinal measures (472-3), which is, of course, important. Rather, I mean that whether it is a behavioural or an attitudinal measure it should satisfy the above definition of cooperation (which for an attitude would refer to an imagined cost-benefit structure). • (e) I suggest that the authors look again at their use of the terms ‘cooperation’ and ‘altruism’, and their derivatives, in light of these comments. At the same time it would be useful also to check use of the terms ‘prosociality’ and ‘collective action’. ‘Prosociality’ refers to a social action providing a benefit to the recipient and therefore includes both altruism and cooperation, as defined above. I would like to see a

3

definition in the paper of ‘collective action’ since it is used a number of times. My next point argues, more generally, for the importance of clear definitions and distinctions between the different types of prosocial behaviour for this field of research.

4. Following on from comment (e), it seems to me that the discrepant findings in the literature about the effect of environmental harshness on prosociality (the authors refer to this at 42-6) may be partly a function of the literature not distinguishing carefully enough between different types of social interaction, with potentially different cost-benefit structures for the players; broadly cooperation versus altruism.

I declare an interest here. I have a paper (Lazarus 2017, and included with this review), one not cited by the authors, in which I review evidence from humans and other taxa that cooperation (defined, as above, in evolutionary biology) may be an inverted-U function of environmental adversity (i.e. harshness). The review is accompanied by an explanatory evolutionary model that differs from the ideas discussed in the present paper (i.e. selection for present-focussed decision making in harsh conditions). The model is intended to supplement, and not to replace, explanations based on present-future trade-offs. The authors may wish to refer to my paper.

My paper also discusses the point made in the paragraph before last and I quote the passage here (Lazarus 2017, section 6, p. 590):

6. Other forms of prosociality Cooperative behaviour shares motivational and emotional features, and evolutionary and cultural origins, with related prosocial behaviours, beliefs and attitudes, and the influence of adversity has been studied here too. At the level of individual differences those with more adverse life experience exhibit more empathy, compassion and generosity in charitable giving and aid to a stranger (Lim and DeSteno 2016; Lim 2017). The influence of social class on prosociality is controversial, with recent conflicting findings for a range of measures: utilitarianism, empathy, feelings of entitlement, narcissism, theft, lying, cheating, helpfulness, generosity, trust, trustworthiness, volunteering and charitable donation. Some studies find greater prosociality in higher classes (Korndörfer, Egloff, and Schmukle 2015) but others find the opposite class effect (Côté, House, and Willer 2015), to mention just two of the most recent reports. In understanding these apparently conflicting findings it will be helpful to develop more tailored predictions for each measure, as attempted here for

4

cooperation. If the inverted-U relationship proposed here between environmental quality and cooperation should hold also for some of these other forms of prosociality this might resolve some of the (apparent) inconsistencies in the data, since conflicting studies could be situated on opposite sides of the environmental inflection point. As others have noted it is also important to take account of the methodology by which these results have been obtained, from behavioural observations and experiments to self-report, in both the lab and the real world, since they each have their own psychological influences.

I invite the authors to engage with these ideas.

Detailed comments

Title: Since the LHS variables play a major role in the association between environmental harshness and the prosociality variables I suggest that LHS is added to the title.

76. ‘By contrast’ with the previous sentence these references are expected to deal with the effect of the experience of safe conditions during human childhood. However, two seem not to be relevant: [54] is a comparative mammalian paper and [60] is a study of guppies.

121 and 338. Provide references or web links to the two data sets.

123. The countries in the survey: it would be useful, in the supplementary material, to provide lists of the countries in each study uniquely, and in both studies. Country differences may help to explain some differences in the results of the two studies, and will also be useful for comparison with future studies.

134. ‘in’ should read ‘into’.

135-7 & 370-72. ‘In both subsamples, we excluded participants with too many empty cells (more than two standard deviations from their respective subsample mean)’. I find this unclear; ‘empty cells’ seems to refer to missing data whereas the passage in parentheses seems to refer to outliers.

151. ‘parent’s’ should read ‘parents’. ‘end’ should read ‘ends’, both here and in the figures.

5

166. This should probably read ‘health costs are induced’.

177. ‘no’ should read ‘not’.

248 and 403. Discovery and replication samples in the two studies. In the Results section of each study there is analysis of just one set of data. It is not stated whether this is the discovery sample, the replication sample, or the two combined. If it is one of the two samples then the second should be added, or comparison made with the first. If it is the two combined then any differences between the two samples should be described.

255-9. Additional analysis with imputed data, to deal with the problem of missing responses. ‘The results of this analysis are consistent with the results obtained using listwise deletions (see S1 Text in the Supporting Information)’ (257-9). In contrast to this statement in the paper there is an important difference in the results of the two analyses. In the main text analysis the authors conclude from a significant negative association between childhood harshness and collective action (309 and Figure 2) that ‘This implies that a harsher childhood environment is associated with less investment in collective actions later in life’ (309-10). However, the same conclusion is drawn from the additional analysis although now the same correlation is significant and positive (S1 Text, p. 3, lines 3-4 and Figure S1). This requires Discussion in the main text, particularly since it represents a core prediction of the paper.

282-7 and S1 Text. Discussion of the effect of the remaining independent variable, parental education, is missing. Is this because of the standardisation procedure?

291. There seem to be words missing at the end of this line.

29. State what the ‘expected way’ is.

303 & 444. ‘bold arrow’ would probably be better as ‘continuous arrow’, in contrast to the dotted arrows. Certainly in the typescript one cannot tell bold from normal lines.

364-7. Excluding respondents without children. Further explanation would be useful here since the variable ‘only relevant for respondents with children’ (365-6), age at first birth, is not used in this second study (but is in the first). So

6

were childless respondents excluded so that the two studies would be more similar in their respondent characteristics?

390. Insert ‘respectively’ after ‘study 1’?

480. ‘whether people cooperate less under harsher conditions . . .’?

517. ‘administrating’ should read ‘administering’.

531. Delete ‘shocks’?

537-9. Either ‘correlates’ should read ‘correlated’ if the correlation was within the study or a reference should be given if it was a correlation with another study.

560 ‘Their model’ would read better as ‘The model’.

830. Add journal name.

Reference

Lazarus, J. (2017) Cooperation in adversity: An evolutionary approach. Global Discourse 7 (4): 571–598, doi: 10.1080/23269995.2017.1402426

Reviewer #2: This manuscript aims to build on previous research showing a negative association between “environmental harshness” and cooperation in contemporary human societies. Specifically, the authors argue that this association is mediated psychologically by a “present bias” (or future discounting) and is embodied biologically by altering the tradeoff between reproduction and maintenance in favor of reproduction. The manuscript tests this hypothesis through analysis of data from two large, cross-cultural studies: the European Values Study (EVS) and the World Values Survey (WVS).

Overall, the manuscript is clearly written and accessible, both to specialists working in this area as well as a more general scholarly audience. The theoretical and empirical motivations are linked directly to the hypotheses. The steps used to produce the data samples for analysis and the methods of analysis are both well documented. The results of the analysis are linked back to the hypotheses, and the discussion includes a helpful review of some limitations of the study and unresolved theoretical debates.

There are several additional strengths of the research featured in the manuscript. The use of large, cross-cultural surveys presents opportunities to test associations between “environmental harshness” and cooperation in a variety of contexts with measures that arise from a common survey methodology, which should enhance the quality of comparisons. The large sample sizes in these data sets also allow the authors to develop their models using one half of the sample, then validate these models with the other half. The analysis explores direct associations between “environmental harshness” and cooperation, as well as mediations via the trade-off between reproduction and maintenance, which unites two strands of research on environmental harshness that are often pursued separately. The discussion of analytic methods and interpretation of results are also sufficiently nuanced to take into account both the strengths and weaknesses of the data used in the study, which helps clarify the contributions of this research as well as opportunities for future studies to build on them. I also appreciated the choice to focus on “collective action” variables from these particular data sets, rather than moral attitudes about cooperation, which have been the focus of previous research.

Along this these strengths, there are several areas where the manuscript could be improved. I outline these below:

1. The manuscript uses the term “environmental harshness” to refer to a broad range of conditions, defined by the “rate at which external morbidity and mortality causes disability & death in a population” (p.3). The authors go on to explain that this often manifests in contemporary human populations as “economic deprivation and resource scarcity.” I found the use of the term “environmental harshness” to be somewhat confusing in this regard, as I was expecting the research to focus more on ecological conditions rather than social conditions tied to “harshness.” Considering the countries included in these data sets, the real-world challenges reflected in the variables used to operationalize the concept of “harshness” are more directly tied to social conditions and dynamics than they are to ecological conditions and dynamics. The manuscript is fairly clear in how it defines the term “environmental harshness,” so this is not a major concern, but I do wonder if it would be closer to the spirit of the analysis to replace this term with something like “economic deprivation and scarcity,” which probably better reflects the specific measures used in the study and the social contexts where these surveys were conducted.

2. Related to the concern about the definition of “environmental harshness” described above, I felt the manuscript could do more to consider recent research on the evolution of cooperation that shows a positive association between environmental harshness and cooperation. For example, several papers by Smaldino and colleagues (2013a; 2013b; 2013c ) use agent-based models to show that increasing environmental harshness leads to increasing cooperation, as does research by Aktipis, Cronk and colleagues (2011; 2015; 2016; 2019) These positive associations run in the opposite direction as most of the research cited in the manuscript, which is presented as the current consensus from an empirical standpoint. The studies showing positive associations also focus more directly on ecological conditions linked to “harshness,” so perhaps this is one reason they diverge from those cited in this manuscript, which share the focus on economic deprivation and scarcity. Regardless, I would like to see this manuscript engage more directly with this work showing positive associations between environmental harshness and cooperation, both in terms of its theoretical framework and interpretation of its results.

3. The authors suggest one of the manuscript’s primary contributions to research in this field is to test the idea that the association between environmental harshness and cooperation is mediated by a present bias at the psychological level that is in turn embodied at the biological level in terms of reproductive strategies. This is an interesting idea that builds on previous research on variation in human life history strategies in relation to environmental harshness, but there are some limitations in the analysis featured in this manuscript that I think should be discussed. First, the analysis does not examine present bias or other psychological mechanisms directly. Instead, it infers these from specific strategies for reproduction (e.g. early reproduction, more children) and cooperation (e.g. fewer contributions to collective action). These inferences are generally logical, but there may be alternative explanations that don’t involve present bias at the psychological level. For example, research by Amir and colleagues on this topic (2019) has suggested that risk management is an alternative explanation, which is not cited nor discussed in the manuscript. Considering the possibility that reproductive and cooperative strategies may also be shaped by risk management, I feel the authors should address this alternative explanation (and the empirical research supporting it) in their manuscript. Indeed, Amir et al (2018) have empirical evidence that environmental harshness is positively associated with cooperation, connecting this issue to the previous one. Perhaps there are even variables in the EVS and WVS that could be used to test this alternate hypothesis. Also, it’s worth noting that some of the work by this research team also provides support for some of the associations presented in this manuscript (Amir et al 2016)

4. There are several missing details about the EVS and WVS data sets that are important for interpreting the results featured in the manuscript. First, the manuscript mentions the number of countries included in each data set, but does not explain which countries are involved. This is more important for the WVS than the EVS, because part of the justification for including analysis from the WVS was to expand the sample of countries considered beyond Europe. I recommend the authors list these countries included in the samples used for analysis for both data sets, either in the main text or in a table included in the main text, perhaps along with descriptive stats at the country level for each of the main variables. Second, the manuscript should engage with or at least acknowledge the extent of variation in social and ecological contexts that are and are not reflected in these samples. Does environmental harshness (or economic deprivation and scarcity) manifest in similar ways in all countries, or are there variations that should be accounted for? More details along these lines would help develop some key parts of the manuscript where this issue arises. For example, in the discussion, the manuscript notes the fact that “number of children” did not substantially contribute to the construct of the reproduction-maintenance tradeoff in the EVS, but it did in the WVS (p.22). The discussion suggests this may be due to the use of contraceptives and other “cultural factors.” This would be a good opportunity to provide more details about the countries included, specifically variation in demographic profiles and other factors linked to reproduction, as well as factors linked to the measures of environmental harshness and collective action. Finally, the modeling strategy doesn’t appear to account in any way for differences at the country level when estimating the associations among the key variables. A multilevel modeling approach might help clarify the extent that the associations reported in the manuscript are general across all countries.

5. The manuscript’s presentation of results and discussion focuses mostly on the statistical significance of the documented associations, but there is little discussion of effect sizes, aside from a brief mention that the reported effect sizes are generally small (p.14), as well as how the effect sizes for the direct association between environmental harshness and collective action decrease when accounting for mediation via reproductive strategies (p.15). This left me wondering about the substantive significance of these associations in relation to observed variation in the key variables. The bivariate correlation coefficients appear to suggest relatively little co-variance in the two composite collective action measures (volunteering and political action) and the variables representing the two predictor constructs. The large sample sizes used in this study raise the possibility that the statistically significant associations reported are due primarily to the high statistical power enabled by the large sample sizes used in this analysis, rather than effect sizes. More detailed discussion of effect sizes and substantive significance would help balance the interpretation of full significance of these associations.

6. Finally, the Discussion section ends rather abruptly with a discussion of limitations and unresolved issues. I suggest the authors consider including at least a short paragraph or two to return to the main contributions of this work and perhaps suggest some important directions for further research in this area, which would help return to the general motivations of this work.

Addressing these issues should help to highlight the strengths of this manuscript and its valuable contributions to our understanding of the links between environmental harshness, reproductive strategies, and cooperation in contemporary human societies.

Citations

Aktipis, C.A., Cronk, L. & de Aguiar, R. Risk-Pooling and Herd Survival: An Agent-Based Model of a Maasai Gift-Giving System. Hum Ecol 39, 131–140 (2011). https://doi.org/10.1007/s10745-010-9364-9

Aktipis, A., de Aguiar, R., Flaherty, A. et al. Cooperation in an Uncertain World: For the Maasai of East Africa, Need-Based Transfers Outperform Account-Keeping in Volatile Environments. Hum Ecol 44, 353–364 (2016). https://doi.org/10.1007/s10745-016-9823-z

Amir D, Jordan MR, Bribiescas RG (2016) A Longitudinal Assessment of Associations between Adolescent Environment, Adversity Perception, and Economic Status on Fertility and Age of Menarche. PLoS ONE 11(6): e0155883. doi:10.1371/journal. pone.0155883

Dorsa Amir, Matthew R. Jordan, David G. Rand. 2018. An uncertainty management perspective on long-run impacts of adversity: The influence of childhood socioeconomic status on risk, time, and social preferences,

Journal of Experimental Social Psychology, Volume 79, Pages 217-226, ISSN 0022-1031

Amir, D., Jordan, M. R., McAuliffe, K., Valeggia, C. R., Sugiyama, L. S., Bribiescas, R. G., Snodgrass, J. J., & Dunham, Y. (2019). The developmental origins of risk and time preferences across diverse societies. Journal of Experimental Psychology: General.

Cronk L. et al. (2019) Managing Risk Through Cooperation: Need-Based Transfers and Risk Pooling Among the Societies of the Human Generosity Project. In: Lozny L., McGovern T. (eds) Global Perspectives on Long Term Community Resource Management. Studies in Human Ecology and Adaptation, vol 11. Springer, Cham

Yan Hao, Dieter Armbruster, Lee Cronk, C. Athena Aktipis, (2015). Need-based transfers on a network: a model of risk-pooling in ecologically volatile environments, Evolution and Human Behavior, Volume 36, Issue 4, 2015, Pages 265-273, ISSN 1090-5138,

Paul E. Smaldino, Jeffrey C. Schank, and Richard McElreath (2013a), "Increased Costs of Cooperation Help Cooperators in the Long Run.," The American Naturalist 181, no. 4 (April 2013): 451-463.

Smaldino PE, Newson L, Schank JC, Richerson PJ (2013b) Simulating the Evolution of the Human Family: Cooperative Breeding Increases in Harsh Environments. PLOS ONE 8(11): e80753.

Paul E. Smaldino, (2013c). Cooperation in harsh environments and the emergence of spatial patterns. Chaos, Solitons & Fractals, Volume 56, 2013, Pages 6-12, ISSN 0960-0779,

6. PLOS authors have the option to publish the peer review history of their article (what does this mean?). If published, this will include your full peer review and any attached files.

Reviewer #1: Yes: John Lazarus

Reviewer #2: No

---

## [Author Response · Author response to Decision Letter 0]

24 Apr 2020

Subject: Resubmitted Manuscript PONE-D-20-00738 in Draft 

Dear Dr. Capraro, 

Thank you for your helpful review of our manuscript and for providing us with the opportunity to revise and resubmit our work. We wish to thank the two reviewers for their positive evaluation of the work, their helpful comments and advice. In the “response to reviewers” file we provide a point-by-point reply to the reviewers’ comments, with changes to the paper placed in italics. We addressed all the reviewers’ comments and made significant changes to the introduction, clarified several topics and provided additional supportive data. 

We hope you will find this version of the paper acceptable.

With kind regards,

Niels Lettinga, Pierre O. Jacquet, Jean-Baptiste André, Nicolas Baumard, Coralie Chevallier

---

## [Decision Letter · Decision Letter 1]

18 Jun 2020

PONE-D-20-00738R1

Environmental adversity is associated with lower investment in collective actions

PLOS ONE

Dear Dr. Lettinga,

Thank you for submitting your manuscript to PLOS ONE. After careful consideration, we feel that it has merit but does not fully meet PLOS ONE’s publication criteria as it currently stands. Therefore, we invite you to submit a revised version of the manuscript that addresses the points raised during the review process.

We look forward to receiving your revised manuscript.

Kind regards,

Valerio Capraro

Academic Editor

PLOS ONE

Reviewers' comments:

Reviewer's Responses to Questions

**Comments to the Author**

1. If the authors have adequately addressed your comments raised in a previous round of review and you feel that this manuscript is now acceptable for publication, you may indicate that here to bypass the “Comments to the Author” section, enter your conflict of interest statement in the “Confidential to Editor” section, and submit your "Accept" recommendation.

Reviewer #1: (No Response)

Reviewer #2: All comments have been addressed

2. Is the manuscript technically sound, and do the data support the conclusions?

Reviewer #1: Yes

Reviewer #2: Yes

3. Has the statistical analysis been performed appropriately and rigorously? 

Reviewer #1: I Don't Know

Reviewer #2: Yes

4. Have the authors made all data underlying the findings in their manuscript fully available?

Reviewer #1: Yes

Reviewer #2: Yes

5. Is the manuscript presented in an intelligible fashion and written in standard English?

Reviewer #1: Yes

Reviewer #2: Yes

6. Review Comments to the Author

Reviewer #1: Reviewer: John Lazarus

My replies to the authors’ responses are headed by the numbered points from my first review (or by line numbers in the first reviewed paper, following ‘pp’) in square brackets, followed by page number in the Track Changes version of the revised paper. (In their response the authors use the page numbers of the revised clean version but in checking the revision the Track Changes version is preferable.)

I appreciate the authors’ constructive engagement with my comments and their detailed and considered responses. None of my comments are major enough to require another look at the paper.

Looking at this again I see that figures 1 and 3 are redundant and could be removed; figures 2 and 4 repeat them, respectively, while adding the statistics.

[3] 98-103. Definition of cooperation. Now that the authors provide a source (West et al., references 61, 62) for their definition, and explain it, this point is satisfactorily resolved.

[3, particularly 3a, b, e] 136-7, 236-49. The authors are now explicit about the potential delayed benefits of volunteering and political action (with supporting references), and have defined ‘collective action’. So this is OK.

[4] ‘the discrepant findings in the literature about the effect of environmental harshness on prosociality (the authors refer to this at 42-6) may be partly a function of the literature not distinguishing carefully enough between different types of social interaction’. This is a general point that I invited the authors to engage with but it is not essential for their paper that they do so. Although they did not take up this point they did add an interesting discussion (54-83) of a second point I raised here that, ‘. . . it is also important to take account of the methodology by which these results have been obtained, from behavioural observations and experiments to self-report, in both the lab and the real world, since they each have their own psychological influences.’

Detailed comments on first submission

[pp 255-9] 392-413. The new analysis and comparisons between existing analyses in response to my comment are a useful addition to the paper. However, the authors should state at 396-9 whether the new analysis is (a) on the replication or the discovery sample, and (b) on the listwise deletion or imputed data set.

All other comments have been answered satisfactorily.

Comments on new text & new supplementary material

57, 60. ‘correlate’ not the best word here since causation is proposed.

310-12 & S1 Text. Largely ‘consistent with the replication sample’ yes, but point out that there are two additional significant associations with Childhood adversity in the discovery sample and name them.

393, 399, 405, 412, 560, 564. To stick to the variable name in the studies, as elsewhere in the paper, ‘cooperation’ should read ‘collective action’.

456-7. Please explain the logic of ‘because not having children now does not mean that you don’t want children in the future’.

634. ‘linear’ should read ‘monotonic’.

707. ‘contest’ should read ‘context’; ‘are’ should read ‘is’.

687-707. With reference to this discussion on uncertainty the authors might be interested in the following theoretical paper, co-authored by myself, but there is no need to cite it here: Andras, P., Lazarus, J., & Roberts, G. (2007). Environmental adversity and uncertainty favour cooperation. BMC Evolutionary Biology, 7, 240. doi: 10.1186/1471-2148-7-240.

Reference 1 has a subtitle, to be added: ‘The impact of harsh versus unpredictable environments on the evolution and development of life history strategies’.

References 61 & 62. The journal names in the doi of each reference have been switched.

Reference 133 is perhaps incomplete.

S1 Text (and check other text and tables for same comment). As I’m sure the authors know, p values aren’t actually zero, statistics packages merely print them this way; ‘0.00’ should read ‘<0.005’.

Reviewer #2: In addition to the strengths of the initial submission, my review highlighted six issues for improvement. I have read the authors’ reply to each issue and reviewed the relevant changes made to the revised manuscript. I am satisfied with the responses and revisions and support publication of this manuscript. I offer more specific replies on each point below:

1. Use of the term “environmental harshness.”

- The authors have replaced the term “environmental harshness” with “environmental adversity.” I support this change and feel it helps clarify the specific social and environmental conditions that are the focus of this manuscript in relation to variation in reproductive strategies and cooperation.

2. Positive or negative association between environmental adversity and cooperation?

- The revised manuscript has an expanded review of published literature on this question and effectively highlights studies that find both positive and negative associations. I particularly like the way the authors explore possible methodological factors that might help explain these diverging results and link this idea to the results featured in their manuscript. My only suggestions for improving this aspect of the manuscript further would be: a) to reflect on and explore possible differences between self-reported measures of cooperation and behavioral measures (especially via economic experiments or “games”) and b) to add a note explaining that several of the studies cited as support for a positive association between environmental adversity and cooperation use agent-based models as a methodology, which is distinct in important ways from both self-reported and behavioral economics-based measures.

3. Alternative explanations/mechanisms underlying associations between environmental adversity and present bias.

- The revised manuscript features an extended discussion of alternative explanations/mechanisms, including uncertainty management. I appreciate the way the authors consider these alternatives and discuss how the results in this manuscript do and do not align with them.

4. More information on EVS and WVS data sets.

- The supplementary files associated with the manuscript now provide more detailed data about the countries included in the EVS and WVS data sets, as well as descriptive statistics for the variables used in analysis. I did notice one possible issue (hopefully minor) in Table S1 for the EVS, the range for the “Age1stBirth” variable goes from 12-83. My understanding is that this variable represents the age when each person responding to the survey first became a parent. Does this mean that the oldest parent in the data set was 83 years old when their first child was born? This would be very old for a male (but not unprecedented) and would, as far as I can tell, be a world record for a female. If the authors have not already looked into this, it might be good to double check the extreme old/young values for this variable in relation to other variables (gender, age) to make sure there are no errors in this important variable.

5. Effect sizes for SEMs.

- The authors have added some additional information about effect sizes for the associations reported in the SEMs and include some discussion of effect sizes in their assessment of the relationships between environmental adversity, reproduction, and cooperation. The manuscript’s conclusions continue to emphasize results of statistical significance tests the direction of associations, but given the large sample sizes, I still feel the discussion of effect size could be expanded to establish links to the substantive significance of these associations. Perhaps this could be done by linking the estimated coefficients to units of measurement for each variable, then discussing what would constitute a meaningful change in levels of cooperation and how large a change in environmental adversity/reproductive strategy is required to produce this change, according to the SEMs. Perhaps the authors may find this recent article useful in this regard:

Brenna Gomer, Ge Jiang & Ke-Hai Yuan (2019) New Effect Size Measures for Structural Equation Modeling, Structural Equation Modeling: A Multidisciplinary Journal, 26:3, 371-389, DOI: 10.1080/10705511.2018.1545231

6. Summary paragraph.

- The revised manuscript contains a summary paragraph at the end of the Discussion section that nicely states the key contributions of this study.

I thank the authors for their considerate responses to the issues raised in my initial review.

7. PLOS authors have the option to publish the peer review history of their article (what does this mean?). If published, this will include your full peer review and any attached files.

Reviewer #1: Yes: John Lazarus

Reviewer #2: No

---

## [Author Response · Author response to Decision Letter 1]

10 Jul 2020

Our response can be found in the file "Response to Reviewers".

---

## [Editor Report · Decision Letter 2]

14 Jul 2020

Environmental adversity is associated with lower investment in collective actions

PONE-D-20-00738R2

Dear Dr. Lettinga,

We’re pleased to inform you that your manuscript has been judged scientifically suitable for publication and will be formally accepted for publication once it meets all outstanding technical requirements.

Kind regards,

Valerio Capraro

Academic Editor

PLOS ONE
---

## [Editor Report · Acceptance letter]

17 Jul 2020

PONE-D-20-00738R2 

Environmental adversity is associated with lower investment in collective actions 

Dear Dr. Lettinga:

I'm pleased to inform you that your manuscript has been deemed suitable for publication in PLOS ONE. Congratulations! Your manuscript is now with our production department. 

Kind regards, 

on behalf of

Dr. Valerio Capraro 

Academic Editor

PLOS ONE